# Simultaneous Determination of Moxifloxacin Hydrochloride and Dexamethasone Sodium Phosphate in Rabbit Ocular Tissues and Plasma by LC-MS/MS: Application for Pharmacokinetics Studies

**DOI:** 10.3390/molecules27227934

**Published:** 2022-11-16

**Authors:** Xinxin Zhao, Yanjuan Yuan, Qing Shao, Hongqun Qiao

**Affiliations:** 1School of Pharmaceutical Sciences, Nanjing Tech University, Nanjing 211816, China; 2Jiangsu Center for Safety Evaluation of Drugs, Jiangsu Provincial Institute of Materia Medica, Nanjing 210009, China

**Keywords:** HPLC-MS/MS, ocular tissue, moxifloxacin hydrochloride, dexamethasone sodium phosphate, ocular infection

## Abstract

Treatment of ocular infection involves pharmacotherapy with steroids and antibiotic drops, such as moxifloxacin hydrochloride (MFH) and dexamethasone sodium phosphate (DSP). To characterize the pharmacokinetics of these two compounds, we performed and validated a liquid chromatography-mass spectrometry (LC-MS/MS) method to quantify them in rabbit ocular tissues and plasma. We used protein precipitation to extract the compounds. The analyte and internal standard (IS) were separated using a Shim-pack Scepter C_18_ column. The mobile phase was composed of 0.1% formic acid water (A) and methanol (B). MFH and DSP were detected using positive ion electrostatic ionization (ESI) in multiple reaction monitoring mode (MRM). The calibration curves for both compounds showed good linearity over concentrations ranging from 0.5 to 200 ng/mL in rabbit ocular tissues and plasma. The lower limit of quantification for both MFH and DSP was 0.5 ng/mL. We validated this method for selectivity, linearity (r2 > 0.99), precision, accuracy, matrix effects, and stability. Thus, we used this method to assess the pharmacokinetic (PK) characteristics of MFH and DSP in rabbit ocular tissues and plasma after single doses. Our results indicate that this method can be used for the simultaneous analysis of moxifloxacin hydrochloride and dexamethasone sodium phosphate in clinical samples.

## 1. Introduction

Bacteria are the leading cause of eye infections worldwide; infections are associated with many factors and can be mono- or multi-microbial [1]. Eye diseases, such as conjunctivitis, keratitis, endophthalmitis, blepharitis, orbital cellulitis, and dacryocystitis, are all caused by bacterial infection [2]. If left untreated, ocular infections may damage eye structures, resulting in vision impairment and even blindness [3,4]. Eliminating infections and regulating the immune response are crucial stages in the treatment of ocular inflammatory disorders [5,6]. Currently, steroid and antibiotic drops are used in conjunction for efficient control of ocular infections. Such combination treatments enhance patient compliance, improve clinical outcomes, and achieve higher rates of bacterial eradication.

MFH is a broad-spectrum antibiotic that limits cell reproduction by inhibiting the type II topoisomerase DNA gyrase and topoisomerase IV [7,8]. MFH is used to treat bacterial conjunctivitis and keratitis and prevent eye infection following surgery. It has shown potent efficacy against a wide range of Gram-negative and Gram-positive ocular pathogens, including multidrug-resistant strains.

DSP is a glucocorticoid that is mainly used as an anti-inflammatory agent and immunosuppressant [9,10]. Anti-inflammatory medications may reduce retinal deterioration and aid in the maintenance of visual function. In the treatment of infectious endophthalmitis, the goal of anti-inflammatory therapy is to reduce intracellular cytokine toxicity, decrease the production of inflammatory cells and antigens following antibiotic delivery, and reduce tissue damage caused by infiltrating leukocytes [11,12]. The chemical structures of MFH and DSP are shown in Figure 1.

Several in vitro assay methods for the simultaneous bioanalytical quantization of MFH and DSP have been reported, and an RP-HPLC method has been proposed for the quantitative determination of MFH and DSP in ophthalmic suspensions [13,14]. However, the sensitivity, speed, and throughput requirements for biosample analysis may not be satisfied by these approaches. Additionally, the PK characteristics of MFH and DSP in biological matrices, such as plasma and urine, have been evaluated in humans and monkeys using several analytical and bioanalytical approaches [15,16]. However, to the best of our knowledge, no report has been published on the simultaneous bioanalytical measurement of MFH and DSP in the ocular tissue matrix using LC-MS/MS. The PK properties of MFH and DSP in rabbits, humans, and monkeys have been previously reported [17,18]. In these in vivo studies, LC/MS or LC-MS/MS was chosen for quantitation in biological samples. These investigations provide us with useful references for our experimental design and analysis. However, these PK studies did not provide detailed information about the analytical process or report the methodology, collected data, or chromatograms.

Hence, herein we propose a quick, selective, and sensitive LC-MS/MS approach with sample preparation protocols for quantifying MFH and DSP concurrently in rabbit ocular tissues and plasma.

## 2. Results and Discussion

### 2.1. Method Development

Tandem MS spectrometric parameters in the ESI source were optimized by a direct infusion of a standard solution of MFH, DSP, and IS using a syringe pump. A stronger and more stable MS signal for MFH, DSP, and IS was observed in positive-ion mode, which was finally selected. Various fragmentors were evaluated, and the best conditions for MFH, DSP, and IS were *m*/*z* 402.3 > 384.2, *m*/*z* 393.3 > 373.2, and *m*/*z* 237.1 > 194.1, respectively. The positive parent ion mass spectra and product ion mass spectra of MFH, DSP, and IS are shown in Figure 2. For MFH, the optimized declustering potential and collision energy were 76 and 31 V, respectively. For DSP, the optimized declustering potential and collision energy were 60 and 12 V, respectively. The optimized declustering potential and collision energy for IS were 60 and 27 V, respectively. 

Various LC conditions were tested to obtain MFH, DSP, and IS peaks with adequate retention times and separation. MFH, DSP, and IS were separated on an Agilent 1260 series, which was connected to a Shim-pack Scepter C_18_ column (4.6 mm, 50 mm, 3 μm). To optimize chromatography parameters, we tested several gradients, and a simple gradient from 80% to 20% of methanol during 5.5 min was found to afford adequate separation of MFH, DSP, and IS. The total chromatography time was 9 min, where the column was re-equilibrated during the last 3.4 min. In addition, 0.1% formic acid added into the water produced a higher sensitivity and a better peak shape. Thus, the mobile phase was composed of 0.1% formic acid water (A) and methanol (B). The column temperature, autosampler temperatures, and flow rate were further optimized for the appropriate retention time and peak shape. Finally, the selected column temperature and autosampler temperatures were 20 °C and 4 °C, the flow rate was 0.6 mL/min. The injection volume was 10 µL. To ensure a minimal carryover effect, the rinse mode was set to water and methanol (50:50 *v*/*v*) both before and after aspiration. These conditions resulted in good chromatographic resolution with sharp MFH, DSP, and IS peaks.

In the trial stage of sample precipitation extraction, to obtain the best recovery and overall detection resolution, the volume of IS and the actual tested sample were set at 10 and 50 µL, respectively, and the volume of methanol was set at 150 µL.

### 2.2. Method Validation 

#### 2.2.1. Selectivity and Sensitivity

In Figure 3, we present the typical chromatograms for blank samples of the cornea and aqueous humor (AH) to which MFH and DSP had been added. MFH and DSP were not affected by the presence of the other chemicals and thus were easily distinguishable from one another. These observations confirm the specificity of the developed LC-MS/MS method for analysis of MFH and DSP in rabbit ocular and plasma samples; there was no interference from endogenous substances.

#### 2.2.2. Linearity and Lower Limits of Quantitative Detection

The calibration curves and correlation coefficients (*r*) of MFH and DSP in rabbit ocular tissue and plasma are presented in Table 1. The peaks of MFH and DSP were linear over a concentration range of 0.5–200 ng/mL for MFH and DSP in the cornea and AH. The LLOQ had to meet the following criteria: a signal-to-noise (S/N) ratio above 10 and measurement error not exceeding 20%. Therefore, the lowest concentration of the linear concentration range that satisfied these requirements was assumed to be the LLOQ. The lower limit of quantitation (LLOQ; RSD ± 20%) was determined to be 0.5 ng/mL.

#### 2.2.3. Accuracy and Precision

Data regarding the intra- and inter-day precision of the analysis of cornea and AH samples are shown in Table 2. Both the intra- and inter-day assay results were within the acceptable variability ranges. These values were within acceptable limits (i.e., 20% for LLOQ and 15% for other concentrations), indicating that this MFH and DSP analysis method was reproducible and reliable in rabbit ocular and plasma samples.

#### 2.2.4. Recovery and Matrix Effect

Extraction recovery was determined for quality control (QC) samples from the cornea and AH at three different concentrations (1, 10, and 150 ng/mL). The extraction recoveries were >98.5% for all three QC sample concentrations (Table 3). These results indicate that this extraction method can determine MFH and DSP concentrations in the ocular tissue. The matrix effects (MEs) of the three concentrations ranged from 87.36% to 114.78%, indicating that the MEs were minimal. These results suggest that the simple protein precipitation method is suitable for the efficient extraction of MFH, DSP, and IS from rabbit ocular tissue and plasma.

#### 2.2.5. Stability

The stability of analytes under different circumstances is shown in Table 4. In the present study, we assessed the analyte’s stability in terms of long-term storage, bench top, autosampler, and freeze/thaw conditions using two different QC samples; the precision of the quantification method for the analyte was determined to be within 15%. The values for DSP are within parentheses. MFH and DSP generally exhibited high stability in our study.

### 2.3. Pharmacokinetics in Different Ocular Tissues

In this study, we determined the concentration of MFH and DSP in the cornea and AH after topical application. In Figure 4, we show the obtained concentration–time profile of MFH and DSP in rabbit ocular tissues. The MFH and DSP concentration–time profile in the cornea and AH observed in the present rabbit study is consistent with the profiles observed previously in other ocular tissues [19,20]. Absorption of MFH and DSP into the eye was rapid, with maximal concentrations observed within 0.5 h in these anterior tissues. As seen in Table 5, MFH and DSP demonstrated high ocular penetration in rabbits, as rapid absorption and sustained concentrations were observed in anterior ocular tissues through 24 h after a single administration.

After a single bolus injection, the maximum concentration at peak (C_max_) for MFH and DSP in the AH was 2656.7 ± 840.6 µg/mL and 91.4 ± 18.5 µg/mL, respectively. MFH and DSP levels in aqueous humor were relatively low, which indicated that the penetration of MFH and DSP through the cornea into the AH was difficult. The half-lives of MFH and DSP in the AH were 1.3 h and 1.7 h, respectively. Half of the volume of the anterior chamber is evacuated every 46 min [21,22]. Because the observed half-lives of intracameral MFH and DSP are longer than this, we hypothesized that binding and recycling occur in the anterior chamber, delaying their removal. The rabbit is a commonly used animal model for preclinical ocular PK studies. However, the relatively limited availability of robust ocular PK data from humans complicates any evaluation of the predictive accuracy of PK data from rabbits [23].

### 2.4. Pharmacokinetics in Rabbit Plasma

In this study, we determined the concentration of MFH and DSP in the plasma after topical application. In Figure 5, we show the obtained concentration–time profile of MFH and DSP in rabbit plasma. The C_max_ and AUC_0–24h_ values of MFH in rabbit plasma after a topical administration were 18.2 ± 5.5 ug/L and 48.1 ± 18.5 ug h/L, respectively. The C_max_ and AUC_0–24h_ values of DSP in rabbit plasma after a topical administration were 2.1 ± 0.3 ug/L and 13.2 ± 2.4 ug h/L, respectively. The low systemic levels observed suggest that the incidence of systemic adverse reactions after topical ocular dosing would be extremely low. The LLOQ for MFH and DSP in this present assay method is 0.5 ng/mL in plasma, which meets the requirement for trace-amount determination of MFH and DSP in systemic circulation. 

## 3. Materials and Methods

### 3.1. Materials and Reagents

Nanjing Technology University (Nanjing, China) supplied MFH and DSP eye drops (Lot: MDPD-S20201102; specification: moxifloxacin hydrochloride 25 mg with dexamethasone phosphate 1.25 mg; purity: MFH 98.5%, DSP 99.0%). MFH (Lot: 510140-202002; purity: 87.82%) and DSP (Lot: 0562012; purity: 98.44%) were supplied by the National Institutes for Food and Drug Control (Beijing, China). MERCK (Germany) supplied HPLC-grade methanol (lot number: 10999607909). Formic acid (HPLC grade, Lot: 73C1903RV) was acquired from Anaqua Chemicals Supply (ACS), USA. HPLC-grade water was produced using a Millipore MilliQ system (Bedford, MA, USA).

### 3.2. LC-MS Conditions

#### 3.2.1. Liquid Chromatography

An HPLC system from the Agilent 1260 series was connected to a Shim-pack Scepter C_18_ column (4.6 mm, 50 mm, 3 μm). The mobile phase was composed of 0.1% formic acid water (A) and methanol (B), and a flow rate of 0.6 mL/min was used. The injection volume was 10 µL, and the column and autosampler temperatures were 20 °C and 4 °C, respectively (Table 6). 

#### 3.2.2. Mass Spectrometry

A Sciex API 4000+ mass spectrometer (Applied Biosystems, Canada) was used to analyze MFH and DSP. The settings for ion detection were optimized as follows: ionspray voltage: 5500 V; collision gas: 10.0 psi; curtain gas: 30 psi; ion source gas 1:55 psi; ion source gas 2:60 psi; ion source temperature: 550 °C; collision cell exit potential: 12 V; and entrance potential: 10 V. Analyst 1.6.2 software was used to acquire the data. 

### 3.3. Preparation of Standard Solutions, Calibration, and Quality Control Samples

Precise quantities of MFH and DSP were mixed with 100% ethanol to create stock solutions. Working stocks, generated by diluting the parent stock with methanol and water (50:50, *v*/*v*), were used to prepare calibration standards and faulty QC samples. To create the IS stock solution, Carbamazepine was dissolved in methanol to create an IS stock solution, which was further diluted with methanol to produce a secondary stock with a concentration of 2 ng/mL. All stock solutions were stored at 4 °C and brought to room temperature before use. Furthermore, 50 µL of each appropriate working solution was briefly vortex-mixed with 150 µL of blank rabbit plasma, and this mix was spiked with the working standard to yield calibration standards of 0.5, 1, 2, 5, 10, 20, 50, 100, and 200 ng/mL. Simultaneously, QC samples were prepared as LLOQ-0.5 ng/mL, LQC-1 ng/mL, MQC-10 ng/mL, and HQC-150 ng/mL. 

### 3.4. Sample Preparation 

A simple and rapid protein precipitation method was used to obtain plasma, cornea, and AH samples. In an Eppendorf (EP) tube, 50 µL of the material was combined with 10 µL of IS solution (2 ng/mL), and protein precipitation was performed by adding 150 µL of methanol. The material was then vortex-mixed for 3 min followed by centrifugation for 5 min at 16,000× *g*. Next, 160 µL of supernatant was transferred to a 1.5 mL EP tube, and the sample was centrifuged for 5 min at 16,000× *g*. A 10 µL aliquot of the obtained supernatant was then injected into the LC-MS/MS system for analysis.

### 3.5. Bioanalytical Method Validation

The LC-MS/MS method for MFH and DSP analysis in rabbit ocular tissues and plasma was validated in terms of specificity, linearity, accuracy, precision, matrix effect, recovery, process efficiency, and stability according to the guidelines of the United States Food and Drug Administration [24].

#### 3.5.1. Selectivity

To establish the lack of endogenous matrix influence on MFH, DSP, and IS retention periods, six blank cornea and AH were analyzed from six different rabbits. All samples were treated according to the steps described in Section 3.4.

#### 3.5.2. Sensitivity

The sensitivity of the method was assessed using the signal-to-noise (S/N) response ratios of MFH and DSP in the calibration standards. The S/N response ratios should be greater than 3 and 10 for the bottom and upper limits of quantification, respectively.

#### 3.5.3. Linearity

Using a least-squares linear regression model and weighting factor of 1/x^2^, a calibration curve was generated by plotting the analyte/IS peak area ratio against the nominal concentration of each standard. For each validation run, the calibrators should be within 15% of the nominal concentration and 20% of the LLOQ.

#### 3.5.4. Accuracy and Precision

To determine the accuracy and precision, three separate validations were run over two days. Six samples (n = 6) were used for each concentration, and QC samples including the LLOQ, LQC, MQC, and HQC were evaluated in sets. The ratio (%) of the calculated mean concentration to the nominal concentration was defined as accuracy (% bias). Accuracy was expressed as a percentage of nominal concentration (% bias); precision was calculated as percentage relative standard deviation (% RSD). Tolerances for both parameters were deemed acceptable within ±15%, except for the LOQ, for which they had to be within ±20%.

#### 3.5.5. Recovery and Matrix Effect

The recovery values of MFH, DSP, and IS from rabbit ocular tissues were determined by dividing the mean area ratios of the extracted QC samples by those of the samples at the same concentrations obtained by dosing extracted blank samples with analytes from the working standard solutions. An acceptable recovery limit is indicated by a coefficient of variation of less than 15% between the mean recoveries at the three QC levels.

Six distinct blank samples from the rabbit ocular matrix were collected and spiked with the QC samples in order to study the ME. The ME was calculated using the peak area ratio between the corresponding peak areas and those of the analytes resolved in the mobile phase at equivalent concentrations. An external matrix influence is indicated by a ME value outside of the range of 85% to 115%.

#### 3.5.6. Stability

The stability of the analyte in diverse tissue samples was evaluated by assessing six replicates of the QC samples at two concentrations under various settings. After 10 h of observation at room temperature, the short-term stability of the analyte in the sample was evaluated, and after 36 h of observation in the autosampler at 4 °C, post-processing stability was evaluated. After 56 days of storage in a refrigerator at 20 °C, the long-term stability was tested, and the freeze–thaw stability was evaluated after three separate cycles. In addition, the consistency of whole blood was evaluated by centrifuging whole blood samples that had been kept at room temperature for 1 h prior to testing; the average value of L-QC and H-QC concentration should be within 15% of the actual value.

### 3.6. Pharmacokinetic Applications

#### 3.6.1. Animals

All animal experimental procedures and protocols were approved by the Animal Ethical Committee of Jiangsu Center for Safety Evaluation of Drugs (LL-20210218-05; Nanjing, China). All studies adhered to the guidelines set by the Association for Research in Vision and Ophthalmology (ARVO) Statement from the Use of Animals in Ophthalmic and Vision Research as well as those of the Chinese Animal Administration.

New Zealand white rabbits (2.1–2.7 kg, aged 12–16 weeks) were purchased from Nanjing Pukou District Leif Farm (Production License No. SCXK(NAN)2021-0006). All animal rooms were monitored and maintained, except for minor variations, under a 12 h light–dark cycle, a temperature of 22.7 ± 0.9 °C, and a relative humidity of 60.5 ± 5%. Food and water were supplied ad libitum.

#### 3.6.2. Animal Grouping

Rabbits in the single-dose administration group (n = 36) were randomized into six groups corresponding to a total number of six sampling points at 0, 0.17, 0.5, 1, 6, and 24 h after MFH and DSP administration. At the zero-hour time point, each rabbit received 50 µL of MFH and DSP eye drops into the conjunctival sac of each eye as a single bolus dose via a micropipette (n = 6 rabbits per collection time).

#### 3.6.3. Tissue Extraction

Blood samples (3 mL) were collected from around the heart using a syringe feeding into a heparinized tube, and plasma was isolated from the blood by centrifugation at 3000× *g* for 10 min. It was then stored at −20 °C until analysis.

Animals were then euthanized under anesthesia. Immediately after death, a 150 µL sample of AH was obtained from the treated eye via paracentesis using an intraocular cannula. The AH was transferred to an EP tube, and 450 µL of methanol was added to the tube. The mixture was vortexed for 3 min and then centrifuged at 2500× *g* for 5 min. The supernatant was removed and filtered through a 0.22 mm membrane. The resulting AH sample was stored at −20 °C until testing.

The cornea was then removed at the limbus using surgical scissors and knives after rinsing with physiological saline. After removing excess water by placing on filter paper, the removed corneas were weighed and kept in separate glass tubes. Subsequently, methanol was added, the mixture was vortexed, and it was centrifuged at 2500× *g* for 5 min. For testing, the materials were homogenized and kept at −20 °C.

### 3.7. Data Analysis

PK parameters, including C_max_, T_max_, AUC, and t_1/2_, were calculated by noncompartmental analysis. DAS 2.1.1 was used to analyze the basic PK parameters based on various tissue concentration–time profiles (Chinese Mathematical Pharmacology Professional Committee, Shanghai, China). All data are expressed as the mean ± standard deviation (SD).

## 4. Conclusions

We developed and validated an efficient, repeatable, highly selective, and sensitive LC-MS/MS method that enabled us to quantitatively detect MFH and DSP in rabbit ocular tissue. To analyze a large number of samples, the effectiveness of the procedure was enhanced by using small sample volumes and shorter running times. Compared with previously reported methods, the present approach yielded lower quantification limits for the plasma, cornea, and AH. The LLOQ was 0.5 ng/mL. The pharmacokinetic data confirmed that the developed LC-MS/MS assay was appropriate for pharmacokinetic studies and relevant investigations of MFH and DSP.

## Figures and Tables

**Figure 1 molecules-27-07934-f001:**
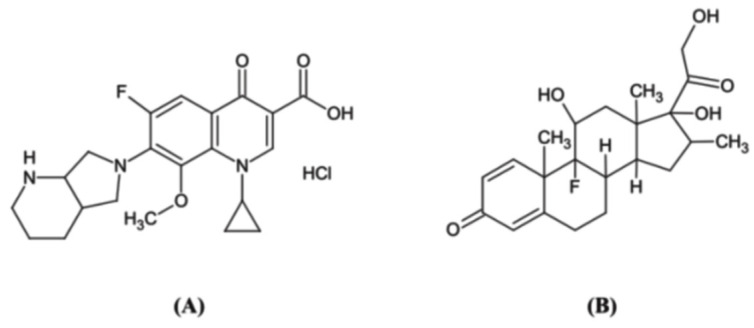
Chemical structures of moxifloxacin hydrochloride (**A**) and dexamethasone sodium phosphate (**B**).

**Figure 2 molecules-27-07934-f002:**
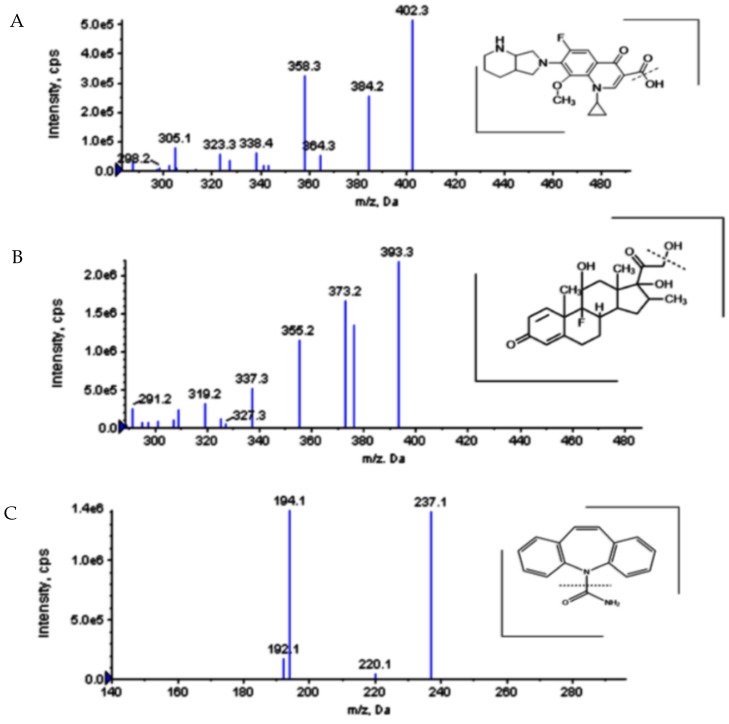
Production spectra (*MS*/*MS*) of moxifloxacin hydrochloride (**A**) (*m*/*z* 402.3 > 384.2), dexamethasone sodium phosphate (**B**) (*m*/*z* 393.3 > 373.2), and internal standard (**C**) (*m*/*z* 237.1 > 194.1) with their possible fragmentation.

**Figure 3 molecules-27-07934-f003:**
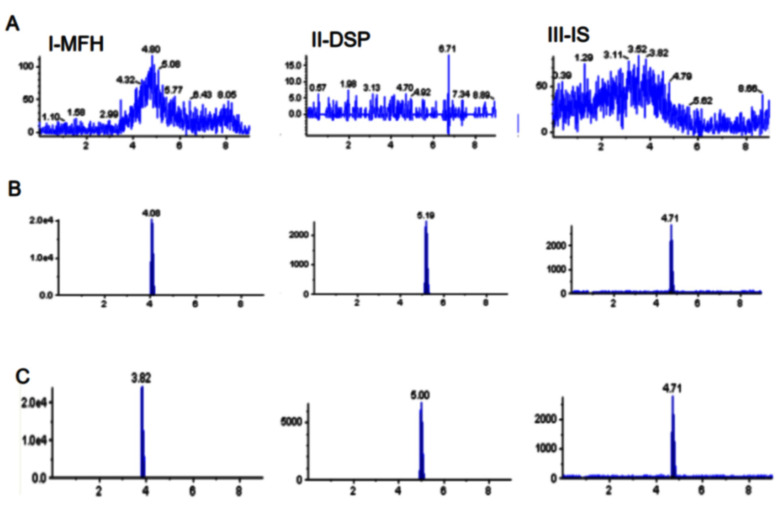
Typical MRM chromatograms of cornea treated with MFH (**A-I**), DSP (**A-II**), and IS (**A-III**) as well as blank cornea spiked with STD6 concentrations of MFH (**B-I**), DSP (**B-II**), and IS (**B-III**). The results of the pharmacokinetic study sample at 10 min are shown in panel C: MFH (**C-I**), DSP (**C-II**), and IS (**C-III**).

**Figure 4 molecules-27-07934-f004:**
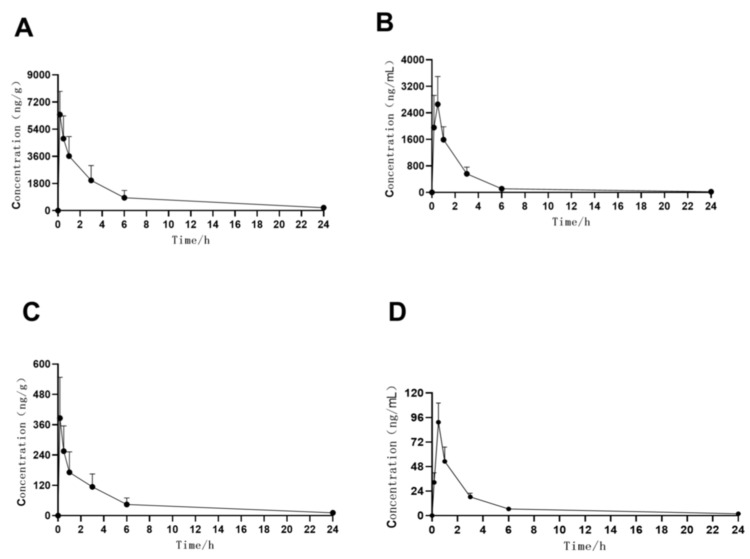
Concentration–time curves of MFH in rabbit cornea (**A**), MFH in rabbit AH (**B**), DSP in rabbit cornea (**C**), and DSP in rabbit AH (**D**) following single topical ocular administration. Data are shown as the mean + SD (n = 6 for cornea, AH). AH, aqueous humor; MFH, moxifloxacin hydrochloride; DSP, dexamethasone sodium phosphate.

**Figure 5 molecules-27-07934-f005:**
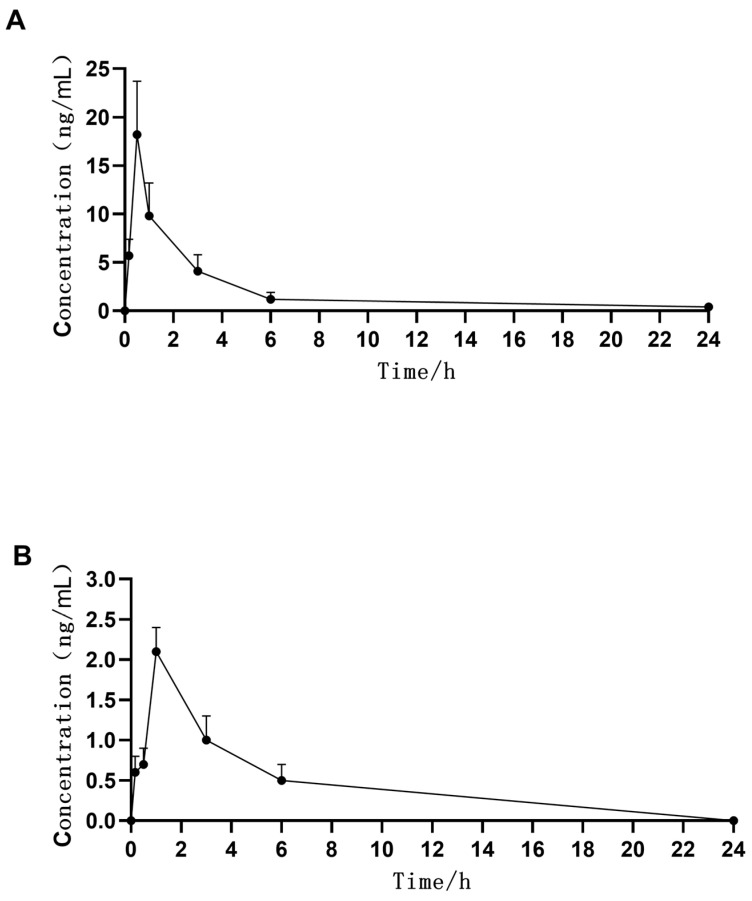
Concentration–time curves of MFH (**A**) and DSP (**B**) in rabbit plasma following single topical ocular administration. Data are shown as the mean + SD (n = 6 for plasma). MFH, moxifloxacin hydrochloride; DSP, dexamethasone sodium phosphate.

**Table 1 molecules-27-07934-t001:** Regression equation and correlation coefficient (*r*) of MFH and DSP in ocular tissue.

	Ocular Tissue	Regression Equation	*r*
MFH	Cornea	Y = 0.506X + 0.0503	0.9973
	Aqueous humor	Y = 0.400X + 0.0212	0.9964
	Plasma	Y = 2.380X + 0.302	0.9983
DSP	Cornea	Y = 0.133X + 0.00434	0.9998
	Aqueous humor	Y = 0.151X + 0.0139	0.9995
	Plasma	Y = 0.161X + 0.0108	0.9997

**Table 2 molecules-27-07934-t002:** Accuracy and precision data for MFH and DSP at the LLOQ, LQC, MQC, and HQC levels in ocular tissue (n = 6).

Ocular Tissue	Concentration (ng/mL)	MFH (DSP)			
		Accuracy (%)		Precision (%)	
		Intra-Day	Inter-Day	Intra-Day	Inter-Day
Cornea	0.5	102.4(107.6)	101.7(103.3)	6.57(4.74)	10.12(9.8)
	1	97.8(109.7)	104.1(111.8)	3.03(6.82)	11.21(7.6)
	10	100.9(95.1)	100.8(97.85)	6.80(4.21)	4.93(4.5)
	150	99.3(91.3)	100.4(96.9)	2.55(3.03)	2.69(6.6)
Aqueous humor	0.5	107.0(97.0)	98.5(92.8)	7.81(10.77)	13.5(11.54)
	1	105.5(100.3)	107(104.88)	4.19(7.05)	12.1(12.9)
	10	103.6(102.7)	102.9(102.5)	6.21(3.57)	4.9(3.23)
	150	101.6(104)	101.8(103.1)	5.73(2.56)	4.1(2.3)
Plasma	0.5	103.6(95.3)	101.2(99.3)	10.7(8.48)	10.2(10.7)
	1	102.1(96.5)	108.2(107.4)	3.01(5.29)	13.5(14.2)
	10	98.8(97.4)	101.6(102.0)	6.02(4.48)	5.7(6.23)
	150	102.6(100.3)	101.3(100.2)	2.83(2.33)	3.1(2.95)

The values for DSP are within parentheses.

**Table 3 molecules-27-07934-t003:** Recovery data for MFH and DSP by HPLC-MS/MS.

Ocular Tissue	Concentration (ng/mL)	MFH(DSP)			
		Recovery (%)	Mean Recovery (%)	Matrix Effect (%)	RSD (%)
Cornea	1	107.2(100.5)	109.0 (99.9)	104.71(95.9)	7.1(11.1)
	10	112.2(105.1)		87.36(101.4)	11.9(9.6)
	150	107.7(94.1)		93.1(95.4)	10.6(7.0)
Aqueous humor	1	98.8(102.2)	98.5 (107.8)	101.92(105.2)	8.4(13.7)
	10	99.6(111.3)		90.6(94.7)	10.4(8.6)
	150	97.1(109.9)		95.59(93.8)	7.5(5.6)
Plasma	1	96.5(103.8)	104.8 (98.7)	114.78(102.4)	10.9(11.8)
	10	109.4(100.1)		90.45(94.9)	12(8.7)
	150	108.5(92.3)		98.84(109.8)	5.8(9.3)

The values for DSP are within parentheses.

**Table 4 molecules-27-07934-t004:** Stability of MFH and DSP under various storage conditions at both the LQC and the HQC (n = 6).

Ocular Tissue	Stability Condition	Nominal Conc. (ng/mL)	Mean	SD	Accuracy (%)
Cornea	Benchtop (room temperature for 10 h)	1	1.0(1.0)	0.1(0.1)	99.4(96.9)
		150	158.3(145.5)	10.5(4.6)	105.6(97.0)
	Autosampler (4 °C for 36 h)	1	1.0(0.9)	0.1(0.0)	103.2(93.2)
		150	159.3(147.7)	10.1(3.5)	106.2(98.4)
	Freeze-thaw (3 cycles; −20 °C to room temperature)	1	1.1(1.12)	0.0(0.1)	109.3(112.3)
		150	158.7(149.3)	8.4(7.8)	105.8(99.6)
	Long-term (−20 °C for 56 days)	1	1.1(1.0)	0.1(0.1)	109.7(103.4)
		150	139.5(142.0)	12.5(6.5)	93.0(94.7)
Aqueous Humor	Benchtop (room temperature for 10 h)	1	1.1(1.1)	0.4(0.3)	107(108.7)
		150	157.5(156.7)	16.7(15.4)	105(104.4)
	Autosampler (4 °C for 36 h)	1	1(1.1)	0.4(0.4)	102(106.2)
		150	150(149.2)	21.2(19.9)	100(99.4)
	Freeze-thaw (3 cycles; −20 °C to room temperature)	1	1.1(1.0)	0.5(0.4)	108.3(103.3)
		150	150.8(147.5)	13.9(11.7)	100.6(98.3)
	Long-term (−20 °C for 56 days)	1	0.9(0.9)	0.4(0.3)	93.3(89.5)
		150	145.8(165.8)	14.3(18.3)	97.2(110.6)
Plasma	Benchtop (room temperature for 10 h)	1	0.9(1.0)	0.4(0.2)	92.8(96.2)
		150	170.8(136.7)	17.2(9.3)	113.9(91.1)
	Autosampler (4 °C for 36 h)	1	1.1(1.1)	0.4(0.3)	109.5(113.5)
		150	146.7(156.7)	27.1(17.8)	97.8(104.4)
	Freeze-thaw (3 cycles;−20 °C to room temperature)	1	1.1(1.0)	0.2(0.3)	112.8(103.5)
		150	151.7(141.7)	23.2(13.3)	101.1(94.4)
	Long-term (−20 °C for 56 days)	1	1.1(1.1)	0.2(0.2)	113.7(111.8)
		150	138.3(154.0)	7.5(15.3)	92.2(102.7)

**Table 5 molecules-27-07934-t005:** Pharmacokinetic parameters.

	Tissue	C_max_	AUC_(0–24h)_	t_1/2_(h)	T_max_ (h)
MFH	Cornea	6369.5 ± 1538.2	23,865 ± 10,777	2.2 ± 0.4	0.2
	Aqueous humor	2656.7 ± 840.6	6311.7 ± 1627.7	1.3 ± 0.2	0.5
	Plasma	18.2 ± 5.5	48.1 ± 18.5	1.6 ± 0.3	0.5
DSP	Cornea	385.8 ± 161.7	1266.7 ± 596.3	2.3 ± 0.6	0.2
	Aqueous humor	91.4 ± 18.5	252.5 ± 47.8	1.7 ± 0.3	0.5
	Plasma	2.1 ± 0.3	13.2 ± 2.4	2.8 ± 1.3	1.0

Units of AH are µg/L and µg h/L, units for cornea ug/g (C) and ug h/g (AUC).

**Table 6 molecules-27-07934-t006:** Instrumental variables in the LC-MS/MS analysis.

Instrumental Variable				
LC	Injection volume	10 µL		
	Column temperature	20 °C		
	Flow rate	0.6 mL/min		
	Mobile phase	0.1% formic acid in water (A) and methanol (B)		
	Gradient change	Time (min)	A%	B%
		0	20	80
		1.5	80	20
		5.5	80	20
		5.6	20	80
		9.0	20	80

## Data Availability

The data presented in this study are available on request from the corresponding author.

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
