# Peer review of "Simultaneous Determination of Moxifloxacin Hydrochloride and Dexamethasone Sodium Phosphate in Rabbit Ocular Tissues and Plasma by LC-MS/MS: Application for Pharmacokinetics Studies"

_molecules, 2022, doi:10.3390/molecules27227934_

Round 1
Reviewer 1 Report
Ethical number for experimental design should be added.
References for validation method should be added.
The result parts of LC-MS, please rewrite.
The resolution of figures should be improved.
The discussion should be improved and cited with updated references.
Author Response
Thank you for giving us the opportunity to submit a revised draft of the manuscript “Simultaneous determination of moxifloxacin hydrochloride and dexamethasone sodium phosphate in rabbit ocular tissues and plasma by LC-MS/MS: Application to pharmacokinetics studies” for publication in the Molecules.We appreciate the time and effort that you and the reviewers dedicated to providing feedback on our manuscript and are grateful for the insightful comments on and valuable improvements to our paper.
We have incorporated most of the suggestions made by the reviewer.Those changes are highlighted within the manuscript.Please see below, in blue, for a point-by-point response to the reviewers’comments and concerns.
Author response: Thank you!
- Ethical number for experimental design should be added.
Response: Thank you for pointing this out. We have updated the summary section.The revised text reads as follows on:
3.6. Pharmacokinetic applications
3.6.1. Animals
All animal experimental procedures and protocols were approved by the Animal Ethical Committee of Jiangsu Center for Safety Evaluation of Drugs(LL-20210218-05; Nanjing, China).
Institutional Review Board: The animanl study protocol was approved by the Animal Ethical Committee of Jiangsu Center for Safety Evaluation of Drugs(KY20-016 and 2021.02).The ethical review approval number is LL-20210218-05.
- References for validation method should be added.
Response: Thank you for pointing this out. The reviewer is correct, and we have updated the summary section.The revised text reads as follows on:
3.5. Bioanalytical method validation
The LC-MS/MS method for MFH and DSP analysis in rabbit ocular tissues and plasma was validated in terms of specificity, linearity, accuracy, precision, matrix effect, recovery, process efficiency, and stability according to the guidelines of the United States Food and Drug Administration[24].
- The result parts of LC-MS, please rewrite.
Response: We agree with the reviewer’s assessment. Accordingly, throughout the manuscript, we have revised.
- Results and Discussion
2.1. Method Development
Tandem MS spectrometric parameters in the ESI source were optimized by a direct infusion of standard solution of MFH,DSP, and IS using a syringe pump.A stronger and more stable MS signal for MFH,DSP, and IS was observed in Positive-ion mode, which was finally selected. Various fragmentors were evaluated and the best conditions for MFH,DSP, and IS were m/z 402.3>384.2, m/z 393.3>373.2,and m/z 237.1>194.1, respectively.The positive parent ion mass spectra and product ion mass spectra of MFH, DSP, and IS are shown in Figure 2. For MFH, the optimized declustering potential, collision energy were 76, and 31 V, respectively, For DSP, the optimized declustering potential, collision energy were 60, and 12 V, respectively, whereas those for IS were 60, and 27 V, respectively.
Various LC conditions were tested to obtain MFH, DSP, and IS peaks with adequate retention times and separation. MFH,DSP, and IS were separated on an Agilent 1260 series was connected to a Shim-pack Scepter C18 column (4.6 mm, 50 mm, 3 μm). To optimize chromatography parameters, we tested several gradients, and as a result, a simple gradient from 80% to 20% of methanol during 5.5 min afforded adequateseparation of MFH, DSP, and IS. Total chromatography time was 9 min, where the column was re-equilibrated during the last 3.4 min.In addition, 0.1% formic acid added into the water produced a higher sensitivity and a better peak shape. Thus, the mobile phase was composed of 0.1% formic acid water (A) and methanol (B), and a flow rate of 0.6 mL/min was used. The injection volume was 10 µL, and the column and autosampler temperatures were 20 °C and 4 °C, respectively. To ensure a minimal carryover effect, the rinse mode was set to water and methanol (50:50 v/v) both before and after aspiration.These conditions resulted in good chromatographic resolution with sharp MFH, DSP, and IS peaks.
In the trial stage of sample precipitation extraction, considering overall the detection resolution, and the volume of IS and the actual tested sample were 10 and 50 µL respectively, we finally set the volume of methanol at 150 µL to get the best recovery and resolution in the total volume of 210µL.
Figure 2. Production spectra (MS/MS) of moxifloxacin hydrochloride (A) (m/z 402.3>384.2),dexamethasone sodium phosphate (B) (m/z 393.3>373.2), and internal standard (C) (m/z 237.1>194.1), with their possible fragmentation.
2.2. Method Validation
2.2.1. Selectivity and sensitivity
In Figure 3, we present the typical chromatograms for the blank samples of the cornea and aqueous humor (AH), to which MFH and DSP had been added. MFH and DSP were not affected by the presence of the other chemicals and thus were easily distinguishable from one another. These observations confirm the specificity of the developed LC-MS/MS method for MFH and DSP analysis in rabbit ocular and plasma without interference from endogenous substances.
Figure 3. Typical MRM chromatograms of cornea treated with MFH (A-I), DSP (A-II), and IS (A-III), and blank cornea spiked with STD6 concentrations of MFH (B-I), DSP (B-II), and IS (B-III). The results of the pharmacokinetic study sample at 10 minutes are shown in panel C: MFH (C-I), DSP (C-II), IS (C-III).
2.2.2. Linearity and Lower Limits of Quantitative Detection
The calibration curves, correlation coefficients(r) of MFH and DSP in rabbit ocular tissue and plasma were presented in Table 1.The peaks of MFH and DSP were linear over a concentration range of 0.5–200 ng/mL for MFH and DSP in the cornea and AH. The LLOQ had to meet the following criteria: the signal-to-noise (S/N) ratio above 10 and measurement error not exceeding 20%. Therefore, the lowest concentration of the linear concentration range that satisfied these requirements was assumed to be the LLOQ. The lower limit of quantitation (LLOQ; RSD±20%) was determined to be 0.5 ng/mL.
2.2.3. Accuracy and precision
The intra- and inter-day precision data for the cornea and AH samples are shown in Table 2. Both the intra- and inter-day assay results were within the acceptable variability ranges.These values were within acceptable limits(i.e., 20% for LLOQ and 15% for other concentrations), indicating that MFH and DSP analysis method was reproducible and reliable in rabbit ocular and plasma.
2.2.4. Recovery and matrix effect
Extraction recovery was determined for quality control (QC) samples from the cornea and AH at three different concentrations (1, 10, and 150 ng/mL). The extraction recoveries were >98.5% for all three QC sample concentrations (Table 3). These results indicate that this extraction method can determine MFH and DSP concentrations in the ocular tissue. The matrix effects (MEs) of the three concentrations ranged between 87.36% and 114.78%, indicating that the MEs were minimal.These results suggest that the simple protein precipitation method is suitable for the efficient extraction of MFH,DSP, and IS from rabbit oculat tissue and plasma.
2.2.5. Stability
The stability of analytes under different circumstances is shown in Table 4. In the present study, we assessed the analyte’s stability in terms of long-term storage, bench top, autosampler, and freeze/thaw conditions using two different QC samples; the precision of the quantification method for the analyte was determined to be within 15%. The values for DSP are within parentheses. In brief, MFH and DSP exhibited good stability in our study.
- The resolution of figures should be improved.
Response:Thank you for pointing this out.Accordingly, throughout the manuscript, we have revised.
- The discussion should be improved and cited with updated references.
Response: We agree with the reviewer’s assessment. Accordingly, throughout the manuscript, we have revised.The revised text reads as follows on:
2.3. Pharmacokinetics in different ocular tissue
In this study, we determined the concentration of MFH and DSP in the cornea and AH after topical application.In Figures 4, we show the obtained concentration–time profile of MFH and DSP in rabbit ocular tissues.The MFH and DSP concentration–time profile in cornea and AH observed in the present rabbit study is consistent with the profiles observed previously in other ocular tissues[19,20]. Absorption of MFH and DSP into the eye was rapid, with maximal concentrations observed within 0.5 h in these anterior tissues. As seen in Table 5, MFH and DSP demonstrated good ocular penetration in rabbits, with rapid absorption and sustained concentrations observed in anterior ocular tissues through 24 h after a single administration.
After a single bolus injection,the maximum concentration at peak (Cmax) for MFH and DSP in the AH was 2656.7±840.6µg/mL and 91.4±18.5 µg/mL,respectively. MFH and DSP levels in aqueous humor were relatively low,which indicated that the penetration of MFH and DSP through the cornea into the AH was difficult.the half-lives of MFH and DSP in the AH were 1.3 h and 1.7 h, respectively. Half of the volume of the anterior chamber is evacuated every 46 minutes[21,22]. Since the observed half-lives of intracameral MFH and DSP are longer than this, we hypothesized that binding and recycling occur in the anterior chamber, delaying their removal. The rabbit is a commonly used animal model for preclinical ocular PK studies. However, the relatively limited availability of robust ocular PK data from humans complicates any evaluation of the predictive accuracy of PK data from rabbits[23].
2.4 Pharmacokinetics in rabbit plasma
In this study, we determined the concentration of MFH and DSP in the plasma after topical application. In Figures 5, we show the obtained concentration–time profile of MFH and DSP in rabbit plasma. Cmax and AUC0-24h values of MFH in rabbit plasma after a topical administration were 18.2±5.5 ug/L and 48.1±18.5 ug h/L, respectively. Cmax and AUC0-24h values of DSP in rabbit plasma after a topical instillation were 2.1±0.3 ug/L and 13.2±2.4 ug h/L, respectively. The low systemic levels observed suggest that the likelihood of the incidence of systemic adverse reactions after topical ocular dosing should be extremely low. The LLOQ for MFH and DSP in this present assay method is 0.5 ng/mL in plasma, which meets the requirement for trace-amount determination of MFH and DSP in systemic circulation.
- Moderate English changes required
Response:Thank you for pointing this out.We have put the manuscript through a Editing Services, which is used to improve the English of the manuscript.
Once again, we thank you for the time you put in reviewing our paper and look forward to meeting your expectations. Since your inputs have been precious, in the eventuality of a publication, we would like to acknowledge your contribution explicitly.
Sincerely,
Qiao

Reviewer 2 Report
The authors established LC-MS/MS method for measuring moxifloxacin hydrochloride and dexamethasone sodium phosphate in rabbit ocular tissues. The authors tested the accuracy and precision, linearity, selectivity, sensitivity, stability, and recovery of the method, indicating the method can be used to measure these two compounds in ocular tissues with good precision. There are some comments as below for authors to consider.
1. Please mention how accuracy and precision in percentage was calculated in Table 2.
2. What is the parameter of the ion spray voltage of MS?
3. How was the sample clean-up of precipitation with methanol? Normally, precipitation with methanol can give good extraction efficiency but the sample clean-up is less efficient, and the supernatant may not be very clear. A clean-up step might be required.
4. It would be helpful to provide a Table where it can show the comparison of the concentration spiked and concentration found in ocular tissue.
5. Did the authors consider performing stress test experiments, such as acid/basic, oxidation, light, and temperature?
6. During the LC-MS method optimization, if the authors consider testing different flow rates, injection temperatures and the ratio of methanol to buffer? These are the major parameters to affect the separation, especially when analyzing degradation products of the drugs.
7. It is also very important to analyze the degradation products of drugs under stress conditions. Especially when using a large number of animal models such as 36 rabbits, it would be not a bad idea to explore more scientific problems.
Round 2
Reviewer 1 Report
Could be published.